

# Spatial ecology of little egret (*Egretta garzetta*) in Hong Kong uncovers preference for commercial fishponds

Chun-chiu Pang[1,2,*], Yik-Hei Sung[3,*], Yun-tak Chung[1], Hak-king Ying[1], Helen Hoi Ning Fong[1] and Yat-tung Yu[1]

[1] The Hong Kong Bird Watching Society, Hong Kong SAR, China
[2] School of Biological Sciences, The University of Hong Kong, Hong Kong SAR, China
[3] Science Unit, Lingnan University, Hong Kong SAR, China
[*] These authors contributed equally to this work.

Corresponding author
Chun-chiu Pang,
pchunchiu@hkbws.org.hk

## ABSTRACT

Many natural wetlands have been converted to human-influenced wetlands. In some instances, human-influenced wetlands could provide complementary habitats for waterbirds, compensating for the loss of natural wetlands. Inner Deep Bay in Hong Kong is composed of both natural and human-influenced wetlands and is under immense development pressure. From an ecology perspective, we need to understand if different wetland types play the same ecological role. To achieve this, we tracked nine little egrets (*Egretta garzetta*) using GPS loggers for 14 months to study their spatial ecology, home range, movement and habitat use. We found that over 88% of the home range of all individuals comprised of wetlands (commercial fishponds, mangrove, *gei wai*, channel, and intertidal mudflat). Among these wetland types, nearly all (seven of nine) individuals preferred commercial fishponds over other habitats in all seasons. Little egrets exhibited seasonal movement and habitat use among seasons, with largest home range, greatest movement, and most frequent visits to commercial fishponds in winter compared to spring and autumn. Our results highlight the significant role of commercial fishponds, providing a feeding ground for little egrets. However, other wetland types cannot be ignored, as they were also used considerably. These findings underscore the importance of maintaining a diversity of wetland types as alternative foraging and breeding habitats.

## INTRODUCTION

In recent centuries, over half of natural wetlands have been lost, and a large proportion have been converted to human-influenced wetlands (*Davidson, 2014*; *Gong et al., 2010*). Such conversion is typically considered detrimental to biodiversity, since many waterbirds rely on natural wetlands as foraging and breeding grounds (*Bellio, Kingsford & Kotagama, 2009*; *Ma et al., 2004*; *Sebastián-González & Green, 2016*). However, some studies have found that human-influenced wetlands could provide alternative, complementary habitats for some species (*Fidorra et al., 2016*; *Giosa, Mammides & Zotos, 2018*; *Kloskowski et al., 2009*;

*Li et al., 2013*; *Márquez-Ferrando et al., 2014*). In some cases, the transformation from natural to human-influenced wetlands has increased bird diversity due to enhanced habitat heterogeneity (*Murillo-Pacheco et al., 2018*). Also, aquaculture commercial fishponds can provide essential feeding grounds for waterbirds (*Navedo et al., 2015*; *Ramirez et al., 2012*). A high number of birds are attracted when commercial fishponds are periodically drained for harvest; the draining practice opens up opportunities for waterbirds, easing the capture of concentrated prey as water depth is reduced (*Navedo et al., 2015*). To understand the ecological role of different wetland types (natural and human-influenced), new studies are needed comparing the suitability of different wetlands to waterbirds, particularly in parts of the world where diverse waterbird communities are being threatened by destruction of wetlands.

The Inner Deep Bay, a Ramsar site in Hong Kong, is an important site for migratory waterbirds, housing over 40,000 birds each winter, including threatened species (*Hong Kong Bird Watching Society, 2018*). The area is a complex landscape with a variety of wetlands (e.g., commercial fishponds, mangrove, and intertidal mudflats) and urban settlements. This area has been under high pressure for housing development (*Morton, 2016*; *Young, 1998*) and wetlands have declined by 53% between 1986 and 2007 (*Ren et al., 2010*). Among wetland types, commercial fishponds are particularly vulnerable because most are located outside the designated Ramsar site and have limited legal protection against development. Data on the habitat use of waterbirds in the area can be used to evaluate the ecological role of different wetland types, thereby providing a basis for wetland conservation and informing land-use management.

Recently, with the technological advancement of tracking devices, tracking studies have been widely used to study the spatial ecology of birds. Advanced tracking methods (e.g., ARGOS or GPS tracking) gather real-time data with accurate location information that traditional bird surveys cannot provide. The resolution of these data can account for variation in movement and habitat use (*Koczur et al., 2018*; *Takano & Haig, 2004*), thereby enhancing our ability to evaluate the habitat quality for birds and yield data to guide habitat management and conservation (e.g., *El-Hacen et al., 2013*; *Mitchell et al., 2016*).

In this study, we used GPS tracking to study the spatial ecology of little egrets (*Egretta garzetta*) in the Inner Deep Bay, Hong Kong. In the area, little egrets are present throughout the year with a population peaks in winter (1,000–2,000 individuals in January) (*Carey et al., 2001*). Since they inhabit a diversity of wetlands, little egrets provide an ideal study system to compare the ecological role of different wetland types (*Martínez-Vilalta et al., 2019*; *Young, 1998*). Further, although the little egret is one of the most widespread ardeid species worldwide (*Martínez-Vilalta et al., 2019*), there has yet been any tracking study investigating its spatial ecology. The main goal of this study was to provide new knowledge on little egret spatial ecology. The specific objectives were to determine home range sizes, movements and habitat use of little egrets. More specifically, we aimed to evaluate whether little egrets exhibit a preference for certain habitats. Since draining of commercial fishponds—usually October—May in the Inner Deep Bay (*Young, 1998*)—drives to reduced water depth and therefore prey concentration (*Navedo et al., 2015*), we expected this seasonal draining to attract little egrets in such period of the year, thus having a major influence on the spatial
ecology of the species. These data may contribute to the conservation of waterbirds in Hong Kong and to guide habitat management in landscape mosaics consisting of natural and human-influenced wetlands worldwide.

## MATERIALS AND METHODS

### Study area

This study was carried out in the Inner Deep Bay of the Hong Kong Special Administrative Region, China (22°29′N 114°02′E). The area consists of a natural, shallow estuarine bay with extensive intertidal mudflats connected to mangroves and human-influenced wetlands, including *gei wais* (tidal shrimp ponds), drainage channels and commercial fishponds. The commercial fishponds form a continuous wetland habitat of approximately 460 ha. Individual fishponds are generally 1–3 hectares in size, and contain polycultures of commercial freshwater fish, including grass carp *Ctenopharyngodon idellus*), grey mullet (*Mugil cephalus*) and tilapia (*Oreochromis* sp.).

### Bird capturing and tracking data collection

From January–December 2018, we captured nine individuals of little egret (*Egretta garzetta*) using clap nets (1.5 m and 2 m in diameter) with fish bait. We put each individual into a covered, large laundry hamper. They are soft enough for the birds from getting hurt, but strong enough for retaining the birds. We attached to each bird a solar-charged GPS-UHF logger [model PICA (5.5 g in weight) or HARRIER (12 g), Ecotone Telemetry, Poland]), using Teflon tape and a backpack harness. The captured birds weighed 290–495 g. The weight of the loggers and harnesses were <3% of the birds' weights. All birds were released within two hours at the site of capture. We programmed the loggers to record data (location and speed) hourly from 5 to 7 pm local time, thus tracking movements from around sunrise (before the egrets leave their roosting sites) to after sunset (when they return to roost). Data were automatically stored on the loggers, and were remotely downloaded every two weeks using a hand-held base station with unidirectional antenna. We included data collected between 30 Jan 2018 and 22 Mar 2019 in the analysis. All procedures were approved by the Agricultural Fisheries and Conservation Department of the Hong Kong Government [permit number: (43) AF GR CON 09/51 Pt. 6, (99) AF GR CON 09/51 Pt. 6, (166) AF GR CON 09/51 Pt. 6, (79) AF GR CON 09/51 Pt. 7].

### Habitat availability

To determine habitat availability in the study area, we first mapped the study area using QGIS 3.6.1 (*QGIS Development Team, 2016*). Next, we delineated and classified the area into six habitat types using Google Earth: commercial fishponds, *gei wais* (tidal shrimp ponds), mangroves, intertidal mudflat, drainage channels and human settlement. Subsequently, we conducted fieldwork to ground-truth the habitat type. Further, we collected the draining schedules of 591 commerical fishponds by interviewing their owners throughout the study period, which covers 81.5% of all commercial fishponds in the entire Deep Bay area.

## Data analysis

We performed all statistical analyses in R (*R Core Team, 2019*), using the packages 'BBMM' and 'adehabitatHR' for home range analysis (*Calenge, 2011*; *Nielson, Sawyer & McDonald, 2013*); 'adehabitatHS' for habitat selection analysis (*Calenge, 2011*); 'lme4' for model fitting (*Bates et al., 2014*) and 'ggplot2' for graphic production (*Wickham, 2016*).

We applied three methods to calculate the home range of individuals. First, given that the location was recorded regularly each hour, we used the Brownian bridge movement model (BBMM) to report the 50% and 95% home range as the core area and overall home range, respectively (*Fischer, Walter & Avery, 2013*). Based on our preliminary field testing, we set 20 m as the location errors for BBMM. We also calculated home range using fixed kernel density estimation (50% and 95% kernel) (*Worton, 1989*) and minimum convex polygon (MCP) (*Mohr, 1947*). For kernel, we used the $h_{ref}$ kernel density estimators (*Calenge, 2011*).

To determine if little egrets exhibit habitat preference, we used compositional analysis, which compares the point habitat occurrence data with habitat availability across the entire home range of each individual (*Aebischer, Robertson & Kenward, 1993*). We defined the 100% minimum convex polygon (MCP) of each individual as their maximum home range, and then calculated the proportion of each habitat type as the habitat availability (*Whisson, Weston & Shannon, 2015*). We then assigned their relocations to the corresponding habitat and calculated the proportion of used habitat. Second, we performed Wilk's Lambda statistic to determine their overall selection of habitat. If preferences were found, we used randomization tests to conduct pair-wise comparisons of resource types (*Aebischer, Robertson & Kenward, 1993*). Consequently, we used the eigenanalysis of selection ratios to examine individual variations in habitat use in different seasons (*Calenge & Dufour, 2006*).

We tested the seasonal effect on daily home ranges, daily travel distance and proportion of daily occurrence in fishponds (i.e., the proportion of GPS fixes on fishponds among all habitats during the daylight period) using Linear Mixed Model (LMM) and Generalized Linear Mixed Model (GLMM) followed by analysis of variance (*Bolker et al., 2009*). For daily home range and daily travel distance, LMMs were applied and fitted with Gaussian distribution. For the proportion of the daily occurrence in fishponds, we constructed GLMMs fitted with binomial distribution and log-linked function. We could only collect data from two individuals in the summer (June to August), so we excluded summer data from this analysis. We set bird identity as a random effect and season as a fixed effect in the model (spring: March to May; autumn: September to November; winter: December to February). Daily home range was calculated as the daily 50% and 95% utilization distribution (UD) of each individual, using the fixed Kernel Density Estimation (KDE) method (*Seaman & Powell, 1996*; *Worton, 1989*). We calculated daily travel distance by summing the travel distance between each successive location on each tracking day. We excluded data collected from 351 tracking days that had missing data.

## RESULTS

Between January 2018 and March 2019, we received 18839 GPS fixes (1296 tracking bird-days) from nine individuals (Table 1). For individuals, the mean ($\pm$ SD) number of tracking days was 154 $\pm$ 41 and GPS fixes was 2,093 $\pm$ 567.

Pang et al. (2020), *PeerJ*, DOI 10.7717/peerj.9893

**Table 1 Home range of little egrets in the Inner Deep Bay, Hong Kong.**

| ID | Tracker model | Start date of tracking | Last date of signal received | Tracking duration (day) | Home range (km$^2$) | | | | |
|----|---------------|------------------------|------------------------------|--------------------------|---------------------|---|---|---|---|
| | | | | | 100% MCP | 50% Kernal | 95% Kernal | 50% BBMM | 95% BBMM |
| CHI01 | PICA | 28/09/18 | 15/11/18 | 49 | 1.54 | 0.13 | 0.79 | 0.14 | 0.85 |
| HUN01 | HARRIER | 01/12/18 | 08/02/19 | 70 | 26.11 | 0.76 | 7.95 | 0.40 | 5.85 |
| HUN02 | HARRIER | 18/10/18 | 22/03/19 | 156 | 12.97 | 1.03 | 5.73 | 0.97 | 6.33 |
| HUN03 | HARRIER | 01/12/18 | 14/02/19 | 76 | 16.16 | 1.53 | 10.28 | 1.37 | 10.17 |
| HUN04 | HARRIER | 01/12/18 | 22/03/19 | 112 | 17.52 | 0.33 | 1.30 | 0.30 | 2.25 |
| PIC05 | PICA | 30/01/18 | 08/02/19 | 217[*] | 38.38 | 3.90 | 24.61 | 2.58 | 20.05 |
| PIC06 | PICA | 30/01/18 | 28/02/18 | 30 | 25.57 | 7.55 | 27.29 | 3.70 | 20.62 |
| PIC07 | PICA | 30/01/18 | 27/02/19 | 394 | 41.04 | 2.03 | 15.32 | 2.20 | 15.81 |
| PIC09 | PICA | 30/01/18 | 21/03/19 | 289[*] | 24.72 | 0.10 | 1.62 | 0.15 | 2.68 |

**Notes.**

[*]PIC05 migrated out of Hong Kong from 13/05/18 to 05/08/18 and PIC09 migrated from 29/03/18 to 01/08/2018. Data obtained in this period were excluded in the analysis.

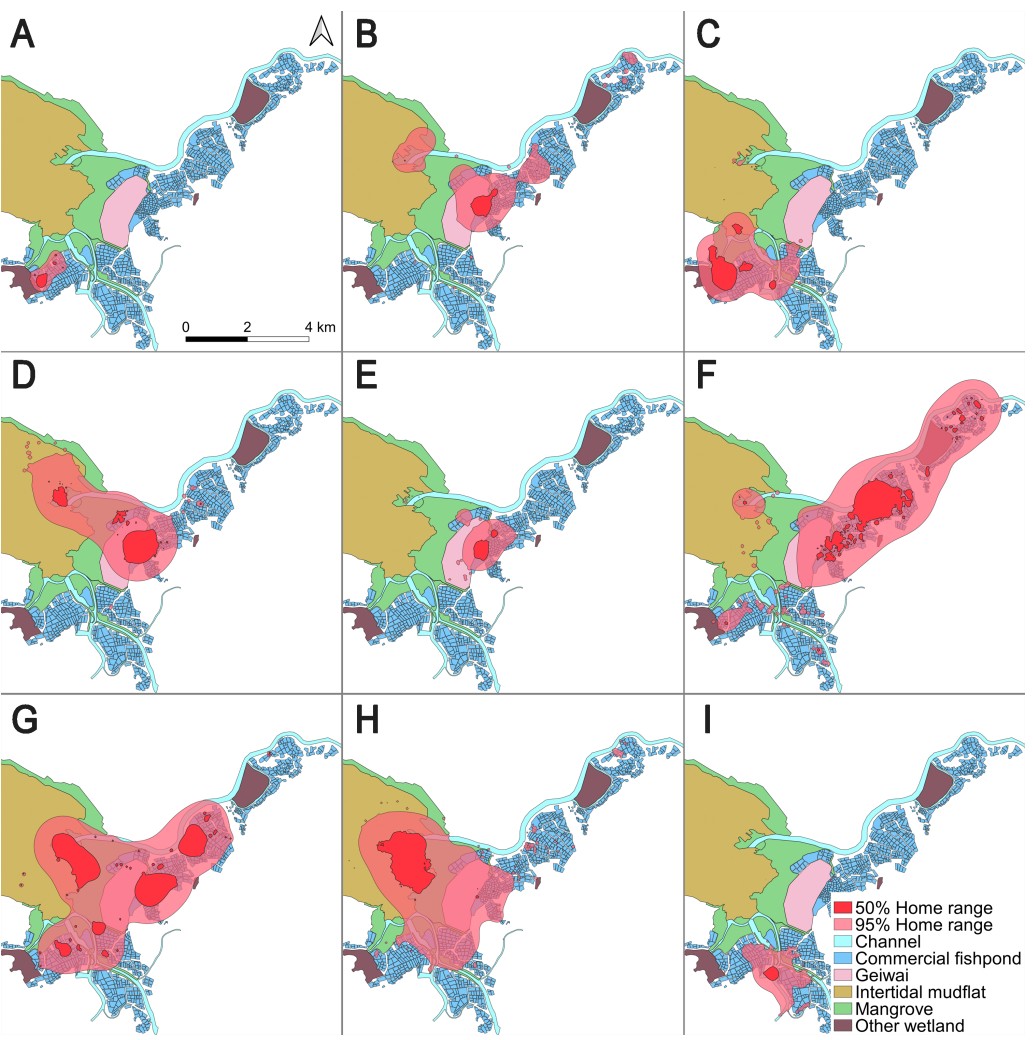

**Figure 1** **50% core areas and 95% overall home ranges of nine little egrets in the Inner Deep Bay, Hong Kong, using Brownian bridge movement model.** (A) CHI01, (B) HUN01, (C) HUN02, (D) HUN03, (E) HUN04, (F) PIC05, (G) PIC06, (H) PIC07, (I) PIC09.

The mean ($\pm$ SD) overall home range (95% BBMM) and core area (50% BBMM) were 9.40 km$^2$ $\pm$ 7.68 (range = 0.85–20.62) and 1.31 km$^2$ $\pm$ 1.26 (range = 0.14–3.70), respectively (Table 1; Fig. 1). The length of the entire tracking duration of each individual did not correlate with the home range size (Pearson's correlation; 50% core area, $t = 0.16$, $df = 7$, $p = 0.87$; 95% home range, $t = 0.48$, $df = 7$, $p = 0.65$).

The home range of all tracked individuals was dominated by fishponds (overall home range = 46.8%; core area = 54.3%), intertidal mudflats (overall home range = 11.4%; core area = 12.5%) and mangroves (overall home range = 13.8%; core area = 7.6%) (Table 2). These three habitat types constituted over 70% of the home ranges of all individuals.

Proportion of habitats used by tracked little egrets (excluding summer) differed from the availability (Wilk's $\lambda = 0.032$, $p < 0.05$ in all cases). Commercial fishponds were the

**Table 2  Proportion (SE) of habitat types within the home ranges for the tracked little egrets, and the proportion of little egret locations within each habitat type in the inner Deep Bay, Hong Kong.**

| Habitat types | 100% MCP | 95% BBMM home range | 50% BBMM home range | Point locations |
|---|---|---|---|---|
| Channel | 6.8 (0.4) | 5.5 (1.6) | 7.5 (5.2) | 9.5 (7.3) |
| Fishpond | 31.3 (2.7) | 46.8 (6.7) | 54.3 (10.4) | 58.7 (9.5) |
| *Gei wai* | 7.4 (1.3) | 11.4 (3.8) | 11.9 (5.7) | 9.7 (4.5) |
| Mangrove | 19.9 (2.5) | 13.8 (2.5) | 7.6 (2.6) | 11.9 (3.4) |
| Intertidal mudflat | 14.0 (2.6) | 11.4 (4.3) | 12.5 (8.3) | 10.2 (5.2) |
| Others | 20.5 (3.9) | 11.1 (3.6) | 6.2 (2.7) | 0.1 (0.0) |

**Table 3  The ranking matrix for habitat selection of the nine little egrets.** The matrix compares the proportion of used habitat based on the relocations and 100% MCP (available habitat); +, preference, −, avoidance, a triple sign indicates significant deviation from random at $p < 0.05$. The ranking list ranges from 0 (most avoided) to 5 (most selected).

| | Fishpond | Mangrove | *Gei wai* | Channel | Intertidal mudflat | Others | Rank |
|---|---|---|---|---|---|---|---|
| Fishpond | | +++ | +++ | +++ | +++ | +++ | 5 |
| Mangrove | −−−−−− | | + | + | + | +++ | 4 |
| *Gei wai* | −−−−−− | −− | | + | + | +++ | 3 |
| Channel | −−−−−− | −− | −− | | + | +++ | 2 |
| Intertidal mudflat | −−−−−− | −− | −− | −− | | +++ | 1 |
| Others | −−−−−− | −−−−−− | −−−−−− | −−−−−− | −−−−−− | | 0 |

most preferred habitats, followed by mangrove, *gei wais*, channel and intertidal mudflat, in preferential order (Table 3). Non-wetland habitats (categorized as 'Others') were the least utilized. However, the eigenanalysis of selection ratios showed individual variation in habitat preference across seasons (Fig. 2). The first two axes explained 87.0% (spring), 100% (summer), 87.6% (autumn) and 86.9% (winter) of the information. Seven of the nine individuals preferred fishponds across all seasons (Fig. 2).

With data from summer excluded, the size of daily 50% home ranges ($F = 67.5$, $df = 2$, $p < 0.001$), daily 95% home ranges ($F = 73.8$, $df = 2$, $p < 0.001$), daily travel distance ($F = 85.0$, $df = 2$, $p < 0.001$), and the proportion of daily occurrence in fishponds ($F = 43.2$, $df = 2$, $p < 0.001$) differed between seasons (Fig. 3, Table 4, Table S1). During winter, little egrets displayed the greatest movement, with largest home range and longest traveling distance; they also visited fishponds more often. The activities declined in spring and reached minimum levels in autumn.

## DISCUSSION

In this study, we examined the spatial ecology of little egrets in the Inner Deep Bay, a complex landscape with a variety of wetlands and urban settlements. We found that little egrets rarely utilized non-wetland habitats (0.1% of all point locations, Table 2), indicating the species is a wetland specialist in the area. We found that little egrets selected

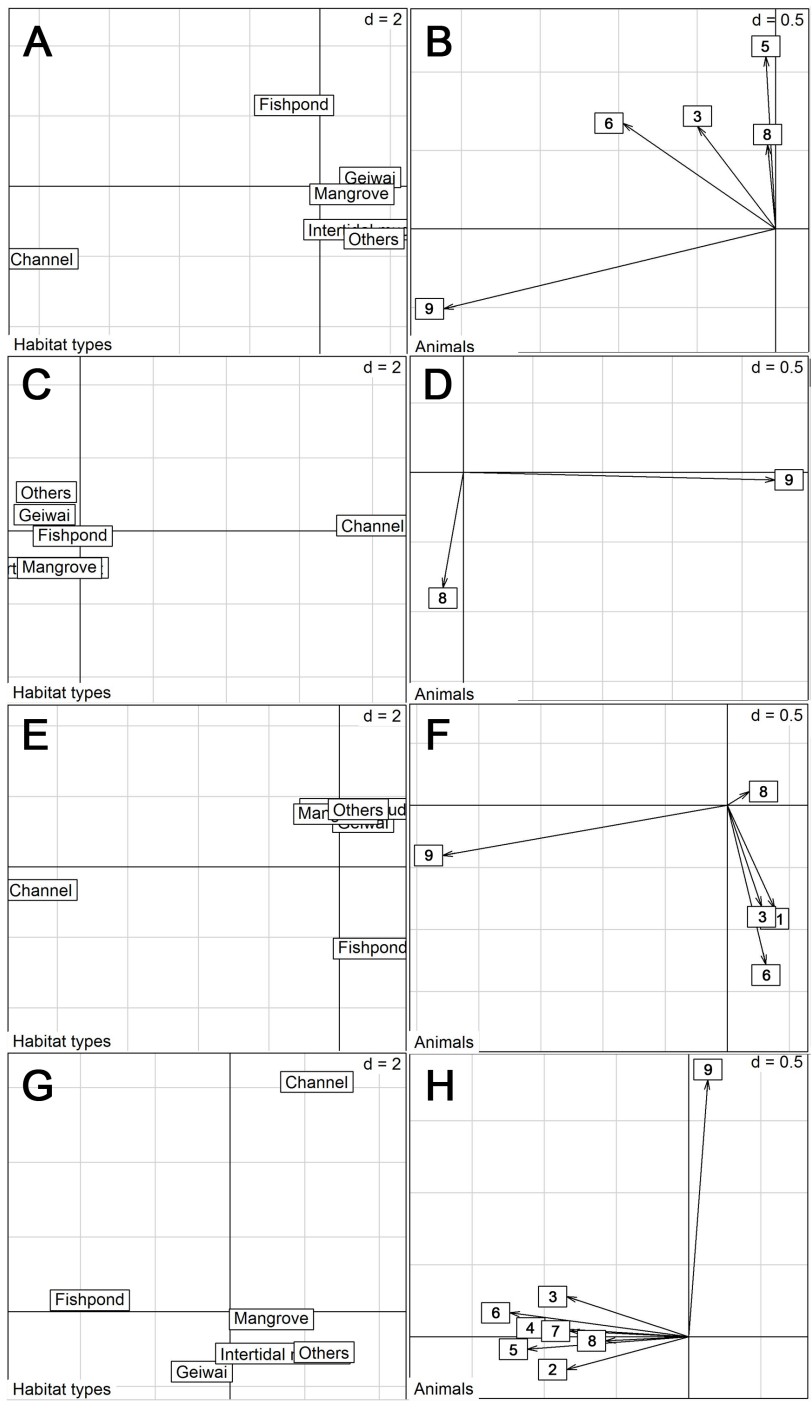

**Figure 2 Eigen analyses of selection ratios of habitat selection of nine little egrets in six habitat types in different seasons.** Habitat types loadings on the first two factorial axes and individual scores on the first factorial plant were displayed by seasons. (A–B), Spring; (C–D), Summer; (E–F), Autumn; (G–H), Winter. The numbers correspond to the animals. 1, CHI01; 2, HUN01; 3, HUN02; 4, HUN03; 5, HUN04; 6, PIC05; 7, PIC06; 8, PIC07; 9, PIC09.

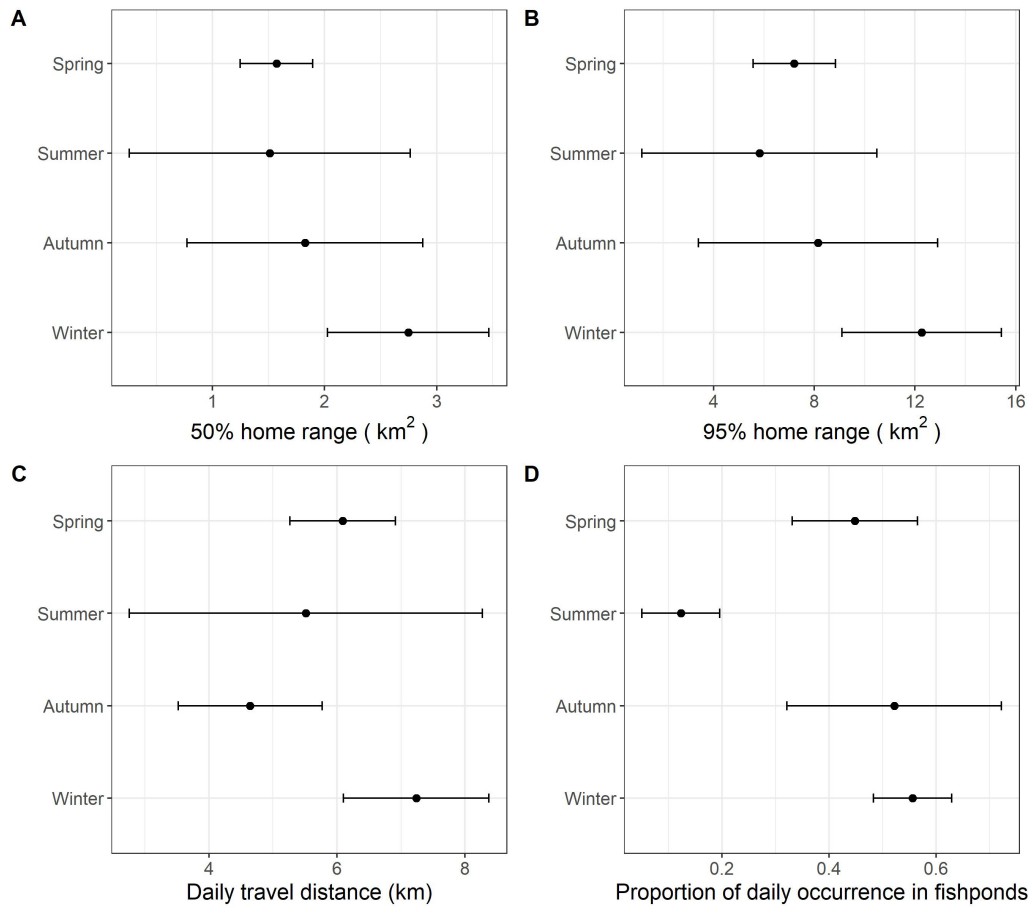

**Figure 3** **The grand mean and standard error of the activities of nine little egrets.** (A) 50% home range, (B) 95% home range, (C) daily travel distances and (D) proportion of daily occurrence in fishponds. Data collected in summer was only visualized but excluded in the analysis due to limited sample size.

habitats nonrandomly and preferred fishponds, and they displayed seasonal differences in movement and habitat use.

## Preference of fishponds

The preference of the little egret for commercial fishponds in the inner Deep Bay agrees with other studies on ardeids (*Fidorra et al., 2016*). The preference of commercial fishponds is probably associated with the draining practices that enhance food availability and accessibility (*Fidorra et al., 2016*; *Rocha et al., 2017*). In our study area, the fishponds are drained for fish harvesting, between October and May (Fig. 4). In drained fishponds, a high density of prey (e.g., fish and invertebrates) become accessible to birds in shallow water (*Young, 1998*). Our data showed that most little egrets rely on fishponds as the major foraging habitat from autumn to spring. Other wetland birds that likely have preference for fishponds, such as the endangered black-faced spoonbill (*Platalea minor*), are often seen feeding alongside little egrets in fishponds (*Yu & Swennen, 2004*). In light of the high
**Table 4 Statistical summary of the regression models for the effects of season on the activities of little egrets.** Summer was excluded in the analysis due to limited sample sizes.

| Source of variation | Model | Season | Estimate | SE | t | p |
|---|---|---|---|---|---|---|
| Log daily 50% home range | LMM | Spring | −0.189 | 0.399 | −0.47 | 0.65 |
| | | Autumn | −1.145 | 0.393 | −2.92 | <0.05 |
| | | Winter | 0.079 | 0.387 | 0.08 | 0.84 |
| Log daily 95% home range | LMM | Spring | 1.354 | 0.395 | 3.43 | <0.01 |
| | | Autumn | 0.345 | 0.388 | 0.89 | 0.40 |
| | | Winter | 1.620 | 0.383 | 4.23 | <0.01 |
| Log daily travel distance | LMM | Spring | 1.637 | 0.163 | 10.0 | <0.01 |
| | | Autumn | 1.154 | 0.160 | 7.21 | <0.01 |
| | | Winter | 1.750 | 0.158 | 11.1 | <0.01 |
| Occurrence frequency in fishponds | GLMM | Spring | 0.494 | 0.096 | 5.17 | <0.01 |
| | | Autumn | 0.459 | 0.095 | 4.84 | <0.01 |
| | | Winter | 0.621 | 0.094 | 6.60 | <0.01 |

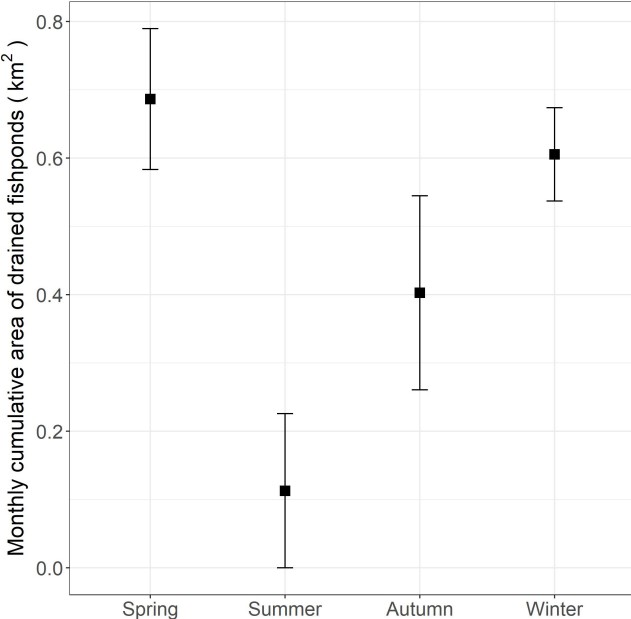

**Figure 4 Monthly cumulative area (and standard errors) of commercial fishponds that were drained according to season in the Inner Deep Bay, Hong Kong in 2018–2019.**

development pressure on fishponds, our findings reinforce the importance of preserving commercial fishponds in the Inner Deep Bay for this group of birds.

## Importance of other wetlands

Despite an obvious preference of commercial fishponds, our data indicate that other wetland types are also important to little egrets as breeding and foraging habitats (>41% of point locations in other wetland types, Table 2). In Hong Kong, little egrets mainly

nest in mangrove, coastal shrubs and trees (*Wong et al., 1999*). One individual (PIC09) shifted from using commercial fishponds in other seasons to mangroves in the summer (breeding season). Overall, the channels, *gei wai*, mangroves and intertidal mudflats were used considerably, as each wetland type held about 10% point locations. Thus, despite fishponds were the major habitat for feeding, other wetland types acted as an alternative feeding grounds to little egrets when there were few drained fishponds available. Although we could not directly test the impact of habitat heterogeneity on movement and habitat use, the use of variety of wetlands by individuals implies wetland heterogeneity is vital to little egrets. Similar studies on other waterbird species would help us understand the impact of habitat heterogeneity on waterbird diversity, in turn informing land managers and governments how to best integrate biodiversity conservation into sustainable development plans.

Moreover, we detected individual variation in habitat use. One individual (PIC07) had distinctive habitat use, preferring a channel over other wetlands (including commercial fishponds) across all seasons (Fig. 2). In the channel that this individual frequented (located at Nam Sang Wai), an ecological-friendly design was implemented, including an unlined earth bottom and mangrove plantation, which has attracted a high number of ardeids and ducks (*Lai, Lee & Wong, 2007*). Overall, our findings also suggest that the coexistence of different wetlands is crucial for accommodating the diversity of individuals and their needs across seasons.

## Seasonal variation in habitat use and movement

We detected significant seasonal differences in habitat use and movement in autumn, winter and spring. Since data were only collected from two individuals in the summer, we were unable to include this period in our analysis. According to our expectation, we found draining schedule of commercial fishponds to influence birds' spatial ecology. The largest home range, greatest movements and highest occurrence in commercial fishponds occurred in winter. We suggest this seasonal pattern to be due to the plentiful, but unpredictable and transient nature of food availability in commercial fishponds. Winter in Hong Kong (December to February) coincides with core of the drainage schedule of commercial fishponds (October to May). Drained commercial fishponds are likely preferred because they contain a large amount of accessible prey. However, food resources in drained fishponds are usually exhausted in a few days (*Rocha et al., 2017*). Searching for resource-rich drained fishponds is probably frequent, but unpredictable (based on fish farmer's preference), which may explain the greatest movements and highest occurrence in commercial fishponds during winter. Conversely, food resources in natural wetlands are likely more predictable.

The monthly cumulative area of fishponds drained in spring in the study period was found comparable to that in winter (Fig. 4), yet we found lower proportion of daily occurrences in fishpond in spring than in winter (Fig. 3), which deviates from our expectation. This hints that movement and habitat use of little egrets are shaped by a balance between foraging and reproduction constraints. In Hong Kong, the reproductive season of little egrets starts in March and April (*Carey et al., 2001*). When they begin

sitting on nests and rearing youngs in mangrove, foraging time and movements get limited (*Maccarone & Brzorad, 2005*), and hence they may not be able to search for and feed in drained fishponds. Tracking more little egrets in summer and areas without fishponds will help elucidate the factors influencing the temporal changes in their habitat use and movement.

## CONCLUSION

Besides improving our understanding of the spatial ecology of little egrets, our results reiterate the importance of preserving wetlands, particularly commercial fishponds, in the Inner Deep Bay in Hong Kong. Human-influenced wetlands can provide not only suitable but preferable habitats for wildlife. Further, the coexistence of different types of wetlands, natural and human-influenced, is important in increasing habitat heterogeneity and providing alternative foraging and breeding habitats for little egrets and other waterbird species. In light of the high development pressure on wetlands in Hong Kong, we hope this study to become a springboard for similar studies to inform us how to better integrate biodiversity conservation into sustainable development plans.

## ACKNOWLEDGEMENTS

We acknowledge the staff of Ecotone Telemetry, particularly A Grochowska, A Komoszyńska and L Iliszko for their technical support on GPS tracking. We thank Dr. Xianji Wen and the staff of World Wide Fund of Hong Kong, especially Banson Leung, in assisting field arrangement at the Mai Po Nature Reserve; Fion Au, Alex Chan and Ivan Tse for field assistance; and many fishpond owners for allowing field work in their ponds. We are grateful to the anonymous reviewers for their careful reading of our manuscript and their many insightful comments and suggestions. We thank Jonathan Fong, Sze-wing Yiu, Amy Fok, Julia Leung and Henry Lee for comments on the manuscript.

### Funding

This study was funded by the Environment and Conservation Fund of the Government of the Hong Kong SAR, China (No. EP 86/19/271). The funders had no role in study design, data collection and analysis, decision to publish, or preparation of the manuscript.

### Grant Disclosures

The following grant information was disclosed by the authors:
Environment and Conservation Fund of the Government of the Hong Kong SAR, China: EP 86/19/271.

### Competing Interests

The authors declare there are no competing interests.

## Author Contributions

- Chun-chiu Pang conceived and designed the experiments, performed the experiments, analyzed the data, prepared figures and/or tables, authored or reviewed drafts of the paper, and approved the final draft.
- Yik-Hei Sung conceived and designed the experiments, analyzed the data, prepared figures and/or tables, authored or reviewed drafts of the paper, and approved the final draft.
- Yun-tak Chung performed the experiments, analyzed the data, prepared figures and/or tables, authored or reviewed drafts of the paper, and approved the final draft.
- Hak-king Ying performed the experiments, authored or reviewed drafts of the paper, and approved the final draft.
- Helen, Hoi Ning Fong performed the experiments, analyzed the data, authored or reviewed drafts of the paper, and approved the final draft.
- Yat-tung Yu conceived and designed the experiments, performed the experiments, authored or reviewed drafts of the paper, and approved the final draft.

## Animal Ethics

The following information was supplied relating to ethical approvals (i.e., approving body and any reference numbers):

Agriculture, Fisheries and Conservation Department of the Government of the Hong Kong SAR, China provided approval for this research [(43) AF GR CON 09/51 Pt. 6, (99) AF GR CON 09/51 Pt. 6, (166) AF GR CON 09/51 Pt. 6, (79) AF GR CON 09/51 Pt. 7].

## Field Study Permissions

The following information was supplied relating to field study approvals (i.e., approving body and any reference numbers):

All procedures were approved by the Agricultural Fisheries and Conservation Department of the Hong Kong Government [permit number: (43) AF GR CON 09/51 Pt. 6, (99) AF GR CON 09/51 Pt. 6, (166) AF GR CON 09/51 Pt. 6, (79) AF GR CON 09/51 Pt. 7].

## Data Availability

The raw GPS relocations of the nine tracked little egrets are available in the Supplementary Files.

## Supplemental Information

Supplemental information for this article can be found online at http://dx.doi.org/10.7717/peerj.9893#supplemental-information.

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
