# Peer review of "Spatial ecology of little egret (Egretta garzetta) in Hong Kong uncovers preference for commercial fishponds"

_PeerJ, doi:10.7717/peerj.9893_

## Round 0.1 · original submission · Major Revisions

Your manuscript has now been assessed by two expert reviewers. Both agree that the work is interesting and could make an important contribution to the literature. However, they both also raised a series of issues that require attention before your manuscript can be considered for publication. In particular, both noted that the study would be considerably strengthened if hypotheses and predictions were incorporated into the introduction to give some much needed context for the study and methods.

From my own reading of the work, I found the introduction quite short, and agree with Reviewer 1 that a more broad background to the rationale and importance of the study would be useful. Reviewer 2 has also provided an annotated copy of the manuscript dealing with several important issues that need clarification. They have also indicated a couple of areas where conclusions drawn appear to be overreaching given the data at hand, notably in the abstract and discussion.

Regarding analysis, please pay particular attention to the sample size issue for summer tracking raised by Rev 1, and I agree with Rev. 2 that you should use your random effect variance estimates to calculate a formal estimate of among-individual variance, rather than using descriptive statistics of the raw data.

There is a fair bit of work required here to address the issues raised, but I've no doubt they will improve the manuscript and make the most of the data you have collected. I look forward to receiving the revised version.

Reviewer 1 ·

Basic reporting

- The manuscript needs to be proof-read to ensure correct language and grammar are used throughout. Currently there are sentences that do not make sense leaving the meaning ambiguous (e.g. line 8, 25, 140). There are also a lot of very short sentences (e.g. line 31) disrupting the flow for the reader

- The introduction provides very little background to the broader relevance and importance of this study, very quickly narrowing in on the specifics of the study system. The first paragraph introduces the current pattern of natural wetland loss/conversion, using broad sweeping statements about impacts/benefits to wildlife. Then, in the second paragraph specific survey techniques are introduced.
The introduction needs to clearly outline the context for this study by discussing the importance of wetland habitats for birds and potential impacts if they are lost. Also, defining natural and human-influenced wetland habitats beyond just examples would be beneficial given the focus on these throughout the paper. Although these two habitat types are highlighted as being distinct the reader can not be sure if both offer equally suitable resources for waterbirds or if there is a cost involved in the conversion. Previous studies have investigated this for numerous waterbird species that rely predominately on human-influenced wetland habitat types (e.g. Richardson et al 2001, Tourenq et al 2001, Sundar 2006, Fidorra et al 2016), I would suggest incorporating this in the introduction as it would highlight the importance of this research

- The author provides very limited information on the ecology of the study species, specifically information on the population of little egrets in this area would provide greater context to the scope of this paper given a small sample size.

- While the discussion section outlines the key findings from the paper it is limited in the depth of the discussion around these points. The author should expand their focus beyond an explanation of the results, addressing broader ecological implications and outcomes.

- There is no discussion around the use of other habitat types beside wetlands despite this being mentioned in the introduction

- Line 16: note that GPS tracking is a type of satellite technology, ‘ARGOS or GPS’ might be a better example.

- Line 179: note it refers to PIC08, this individual is not listed in the study

- Raw GPS data was accessible and in a useable format

- Figure 1: only mentioned in the text in reference to the size of the home range, however individual maps are on different scales making it hard to compare size

Experimental design

- The objectives of this study are clearly defined but repetitive. Objective 2 and 3 largely reiterate Objective 1 just in more detail. Objectives 2 and 3 would benefit from having associated predictions helping the reader focus on the intended outcomes from this paper.

- The use of point fixes in the habitat analysis (Table 2) is potentially problematic with the majority of fixes recorded during winter (Dec- Feb), therefore biasing the result (line 140)

- A sample size of 9 is limiting but not uncommon in studies similar to this. However, I have concern over the analysis approach and specifically seek clarity on the GLMM. Seasonal effect is being tested however there is only n=2 for summer. This needs to be acknowledged in the methods and caveated when drawing conclusions from the results.

Validity of the findings

- There are no test statistics reported from the GLMM and subsequent ANOVA within the text of the results section, despite them being described. They are only referred to in Fig 3 and Table S2, making it hard for the reader to substantiate the results.

- The author has a tendency to make broad sweeping statements from the results that are later contradicted when the data is discussed at a finer scale.
e.g. Line 139 describes a preference for fish ponds across seasons then line 147 notes low occurrence at fish ponds in spring

- The penultimate paragraph in the discussion (lines 177-182) is the first time the author clearly acknowledges the seasonal gap in the data. Given that objective 2 aims to ‘examine seasonal patterns of movement and habitat use’ the author needs to address this earlier as it could have significant ecological implications. Instead the author uses language that alludes to cross seasonal data for all individuals (e.g. line 125/126).
As noted by the author the small sample size over summer also coincides with the breeding season, further highlighting the potential significance of this sampling gap. The paper might benefit from being reframed as a study on the non-breeding habitat selection of foraging little egrets accounting for this. It is problematic for the author to definitively conclude seasonal differences (line 153) with a limited summer sample size.

Additional comments

This study provides a novel contribution to our understanding of the importance of wetlands (natural and human-influenced) for waterbirds such as the little egret in Hong Kong. The conclusion is clear and concise.

Reviewer 2 ·

Basic reporting

In this paper, Pang et al. present novel data about spatial ecology and habitat use of little egrets inhabiting Mai Po Marshes and Inner Deep Bay in Hong-Kong (China). Authors characterize habitat selection and evaluate variation across seasons, discussing the results in the contexts of management of wetland systems. The main novelty of this work comes from the use of GPS tracking with this species. Despite it is abundant and distributed worldwide, so far (to my knowledge) there is no research published providing tracking data of this species. Moreover, results may be relevant for management actions in a wetland system of priority conservation, so I congratulate the authors for that. The manuscript is succinct but the extension is sufficient to provide main results and draw conclusions. However, authors do not present any expectation or hypothesis.

Experimental design

Authors tracked 9 birds with GPS over a whole year recording high-resolution positioning data, used three different approaches to estimate home range, and perform resource selection analysis to evaluate the importance of each kind of habitat for the species. Methods are generally correct, though some clarifications should be provided or analysis corrected, specifically regarding linear mixed models.

Also, authors get into inter-individual differences in habitat use. Yet, to do so they merely discuss it from standard deviation of metrics calculated. As they use (G)LMM, it is possible to provide some more statistical support to such differences taking advantage of random effects outputs.

I provide more in-deep comments in this regards in the attached reviewed document.

Validity of the findings

Statistical methods used looks appropriate, though some clarifications should be done. Findings of the article are useful for conservation purposes, as they can provide guidance in management schedules in human-influenced wetlands. However, authors do not fully take advantage of results during the discussion to highlight such usefulness in the context of wetland management and conservation.

Additional comments

I have some general comments that may help to improve the manuscript and its value for the research community. First, since authors discuss the results within the framework of conservation and management of wetland habitats, I do think introduction could be improved in a more comprehensive way in that regard. Specifically, developing the text (and providing supporting references) indicating that some wetland systems (as it is the case of Mai Po), despite being all wetlands, contains a high diversity of micro- and meso-habitats and that in some cases human alteration/management of these systems may lead to increase such habitat diversity. It is once within this framework where measuring habitat-type selection spatially and over time, and ranking habitat-type use gains value, as you can use the knowledge gained to provide guidance in management. I feel this link is lacking in the text and clearly would improve the work.

Also, I have the feeling that the work would gain in robustness presenting expectations/hypothesis in the introduction. Indeed, the authors present hypotheses as two possible explanations of bird behaviour in the discussion. But these hypotheses could flow directly from habitat diversity of wetland system plus previous knowledge of the system. That is indeed a strength of this work in relation to habitat diversity, habitat use, and the lessons for management. If authors were familiar with the system studied in this work, and previously they knew that fishponds are drained in a specific time window annually -which promotes not only diversity but dynamism- they could make some expectations beforehand regarding spatial ecology of egrets. Doing so, the results would gain relevance as (1) enhance the importance of high-resolution remote-tracking to monitor individual use of habitat in complex systems at several temporal scales and (2) provide robust support to consider a possible impact (positive or negative) on bird community by habitat alteration or by modifying/shifting/quitting management tasks in human-influenced wetlands. This should be reflected in discussion obviously, enriching it. You may want to take these advices into consideration and make some extra-work in the text to improve it.

Annotated reviews are not available for download in order to protect the identity of reviewers who chose to remain anonymous.

---

## Round 0.2 · Minor Revisions

Your manuscript has now been reassessed by two of the original reviewers, whose comment are appended below. Whilst we all agree the manuscript is much improved, a few issues remain. In particular I'd like to see reviewer 1's comments about presentation of figures addressed, and reviewer 2 has highlighted an opportunity to make the tested hypotheses more clear.

Reviewer 1 ·

Basic reporting

By restructuring and adding the suggested detail to the introduction and discussion the author highlights the ecological relevance of their work, justifying the important role this research can play in the conservation and management of wetland habitats.

The author now provides clear, concise objectives, capitalising on the strength of this work, making the overall focus of the paper more coherent and the discussion of results more robust.

Figure 1: The edits made to this figure have significantly improved it, making the differences in home range size and available habitat types clear to the reader.

Figure 3: This figure is very confusing and missing a level of detail that would be useful for the reader. Firstly, the x axis would benefit from being labelled with the relevant units. Also, presenting the data as a difference from autumn (set as 0) is an unusual approach. Displaying all group means and standard error would make the figure clearer for the reader to interrupt. Lastly, from what I understand in 3C, the model estimates are being presented on the link scale instead of as the raw parameter estimates. I feel this should be back-transformed to avoid visually over representing the seasonal differences.

Figure 4: I am now wondering if the size (area^2) of drained fishponds should be considered for this figure rather than simply number drained. Although I am certain they are correlated, given spatial ecology is the overarching focus of this manuscript it might be more relevant to have it as a measurement of space rather than a count, as this is likely to effect food availability. However, this may be a matter of personal style

Line 104: grammatical issues with sentence

Line 263: grammatical issues with sentence

Experimental design

No comment

Validity of the findings

No comment

Additional comments

I very much enjoyed reading the resubmitted manuscript on the spatial ecology and role of wetlands habitat for little egrets in Inner Deep Bay, Hong Kong. The edits provided by the authors have significantly improved the manuscript. I feel the authors have dealt well with the feedback they received and therefore my suggested improvements are minor.

Reviewer 2 ·

Basic reporting

no comment

Experimental design

no comment

Validity of the findings

no comment

Additional comments

Authors have carried out successfully the changes proposed in the first review. The manuscript has improved substantially, so I congratulate the authors. My unique important comment is that the hypothesis statement could still be improved to make it clearer. I attach the document with some comments and suggestions that will help to improve a little bit that weakness. Authors will also find more suggestions along the document that may help to make text clearer to the readers.

Annotated reviews are not available for download in order to protect the identity of reviewers who chose to remain anonymous.

---

## Round 0.3 · Minor Revisions

Thank you for making the requested changes to the manuscript. Figure 3 in particular is much improved and far more easy to interpret.

I have noted only a couple of minor issues that need rectifying, after which I would be delighted to recommend your manuscript for publication.

1) You haven't included degrees of freedom for your F tests in your new edits. These are vital for interpreting the power of the test and the level of hierarchy in the data at which you are testing

2) Figure 4: Bar plots are not ideal for presenting point estimates, and so I'd prefer to see these as point plots, perhaps even with some estimate of variability or error in measurement of the cumulative area.

I look forward to seeing a revision

---

## Round 0.4 · accepted · Accept

Thank you for clarifying the details of your statistics, and updating the figure to include error bars. I am now pleased to recommend that your manuscript be accepted.